# Incidence of Drug-Resistant Hospital-Associated Gram-Negative Bacterial Infections, the Accompanying Risk Factors, and Clinical Outcomes with Treatment

**DOI:** 10.3390/antibiotics12091425

**Published:** 2023-09-09

**Authors:** Lorina Badger-Emeka, Abdullatif S. Al Rashed, Reem Y. Aljindan, Promise Madu Emeka, Sayed A. Quadri, Hayfa Habes Almutairi

**Affiliations:** 1Department of Biomedical Sciences, College of Medicine, King Faisal University, Al-Ahsa 31982, Saudi Arabia; squadri@kfu.edu.sa; 2Department of Microbiology, College of Medicine, Imam Abdulrahman Bin Faisal University, Dammam 31441, Saudi Arabia; asalrashed@iau.edu.sa (A.S.A.R.);; 3Department of Pharmaceutical Sciences, College of Clinical Pharmacy, King Faisal University, Al-Ahsa 31982, Saudi Arabia; pemeka@kfu.edu.sa; 4Department of Chemistry, College of Science, King Faisal University, Al-Ahsa 31982, Saudi Arabia; halmutairi@kfu.edu.sa

**Keywords:** hospital-associated infections, extensive drug resistance, multidrug resistance, antimicrobials, length of hospital stay

## Abstract

Extensive drug resistance to bacterial infections in hospitalised patients is accompanied by high morbidity and mortality rates due to limited treatment options. This study investigated the clinical outcomes of single and combined antibiotic therapies in extensive (XDR), multidrug-resistant (MDR) and susceptible strains (SS) of hospital-acquired infections (HAIs). Cases of hospital-associated drug-resistant infections (HADRIs) and a few susceptible strains from hospital wards were selected for this study. Bacteria identifications (IDs) and antimicrobial susceptibility tests (ASTs) were performed with a Vitek 2 Compact Automated System. Patients’ treatment types and clinical outcomes were classified as alive improved (AI), alive not improved (ANI), or died. The length of hospital stay (LOHS) was acquired from hospital records. The HAI pathogens were *Acinetobacter baumannii* (28%), *Escherichia coli* (26%), *Klebsiella pneumoniae* (22%), *Klebsiella* (2%) species, *Pseudomonas aeruginosa* (12%), *Proteus mirabilis* (4%), and other *Enterobacteriaceae*. They were MDR (40.59%), XDR (24.75%), carbapenem-resistant *Enterobacteriaceae* (CRE, 21.78%) and susceptible (12%) strains. The treatments were either monotherapy or combined therapy with different outcomes. Monotherapy produced positive significant outcomes with *E. coli* infections, while for *P. aeruginosa,* there were no differences between the number of infections treated with either mono/combined therapies (50% each). Nonetheless, combined therapy had significant effects (*p* < 0.05) as a treatment for *A. baumannii* and *K. pneumoniae* infections. Clinical outcomes and LOHS varied with infecting bacteria. The prevalence of XDR and MDR HAIs was found to be significantly high, with no association with treatment type, LOHS, or outcome.

## 1. Introduction

Globally, hospital-associated infections (HAIs) are a leading cause of high morbidity and mortality [1,2]. There is a healthcare burden from bacterial infections that are resistant to currently available antimicrobials [3]. Difficult-to-treat bacterial infections are a leading cause of treatment failures, with complications that lead to long hospital stays and death [4]. The last decade has largely seen the evolution of multidrug-resistant bacteria (MDR), creating immerse public health concerns around the world [5]. Overall, difficult-to-treat Gram-negative bacteria (GNB) have become highly resistant to currently available antibiotics [6], providing limited options for treatment. Thus, clinicians are currently faced with numerous challenges in the treatment of critically ill patients with infections resulting from MDR-GNB [7]. In addition, antimicrobial resistance in HAIs has led to a huge economic burden on both low- and high-income countries, as well as prolonged hospital stays [8,9]. While it is estimated that there are 700,000 deaths per year due to these bacterial infections globally, this number is predicted to rise to 10 million annually by the year 2050 [10]. It is also estimated that about 7.6% of patients in a regular ward will be affected by HAIs, while 50% of those in Intensive Care Units (ICUs) will be affected [11]. These percentages are reported to vary and could be higher in different regions of the world [9]. Many risk factors are associated with HAIs, some of which are the hospitalisation ward, surgery and its type, as well as the patient’s underlying medical conditions [9].

Additionally, with treatment not being delivered as fast as they are required, treatment approaches have evolved that include combinations and alternative options, all of which have made successful treatments difficult for critically ill patients [7]. Thus, the public health problem of MDR bacterial infections continues with no end in sight. The global health problems due to MDR-GNB infections are not abating, and the difficulties faced by clinicians have led to treatment approaches, such as the use of combined therapeutic measures, amongst other options. On the contrary, the literature on the management of such infections, either as single or combined antimicrobial therapies, remains small [6]. The aim of this study was to evaluate hospital-associated drug-resistant infections (HADRIs) by GNB in hospitalised patients, the clinical outcomes of these applied therapeutic measures, and the length of hospital stay (LOHS) in the region of the study. This was with a view to providing more insight into the current therapeutic options in the management of these bacterial infections in hospitalised patients from different hospital wards.

## 2. Materials and Methods

### 2.1. Study Location and Ethical Considerations

This study was conducted at King Fahd Hospital of the University (KFHU), Al Khobar, Saudi Arabia. Located in the southern region of Saudi Arabia, KFHU was established in 1981 and is affiliated with Imam Abdulrahman Bin Faisal University. With a bed capacity of 440, KFHU is considered the main tertiary hospital in Al Khobar city. The institutional review board (IRB) of Imam Abdulrahim Bin Faisal University, under approval number IRB-PGS-2022-01-271, gave approval for the research.

### 2.2. Study Design and Data Collection

This retrospective cohort study investigated infections of drug-resistant and some antibiotic-susceptible bacterial isolates that originated from hospitalised patients from the following hospital wards: Cardiology, ENT, ER, Family Medicine, General Surgery, ICU, Infectious Diseases, Internal Medicine, Nephrology, Neurology, Neurosurgery, Obstetrics and Gynaecology, Oncology, Orthopaedics, Paediatrics, and Urology. MDR, XDR and extended-spectrum beta-lactamase (ESBL) positive strains were defined according to the standardised international terminologies of the Centres for Disease Control and Prevention (CDC) and European Centre for Disease Prevention and Control (ECDC).

The patients were selected based on the laboratory results of antimicrobial susceptibility of GNB isolates obtained from the records in the Microbiology Laboratory database of KFHU between January 2019 and January 2021. A total of 101 non-repetitive cases of single Gram-negative bacterial strains were selected for investigation from the microbiology laboratory records. The bacterial isolates of the HAIs had been preserved in the hospital −80 °C Microbank freezer, from where they were retrieved and re-cultured by plating them out on MacConkey agar. All resultant plates were cultured aerobically for 24 h at a temperature of 37 °C and transferred to the Department of Medical Microbiology, King Faisal University. The bacteria preservation and retrieval methods were performed according to the guidelines provided by the manufacturers (https://www.pro-lab-direct.com/v/vspfiles/microbank/microbank-wwp-portfolio.pdf, accessed on 19 June 2021). Freshly grown, overnight pure bacterial cultures were used for the confirmation of bacterial IDs and AST in the laboratory of the Microbiology Unit, College of Medicine, King Faisal University, Al Ahsa. Inclusion criteria for the selection of patient cases were that the Gram-negative isolates had to be multidrug (MDR) resistant and from hospitalised patients irrespective of age, gender, and nationality. However, 11 susceptible strains were included for control purposes, except for *A. baumannii,* where no susceptible strains (SS) were available. Excluded from the study were infections of Gram-positive bacteria as well as those from outpatient departments. Based on this initial selection of GNB-HAIs, the obtained microbiological data were merged with the specific patient’s automated medical records in the hospital. The relevant patient information that was extracted was socio-demographic characteristics, associated medical comorbidities, clinical diagnosis, wards where samples were collected, antimicrobial therapies, clinical outcomes, and length of hospital stay.

### 2.3. Identification and Antimicrobial Susceptibility Assay of Bacteria Isolates

One hundred and one bacterial pathogens used for the investigation were given laboratory codes consisting of a letter and a number. Codes with A were *Acinetobacter baumannii* isolates, K for *Klebsiella* species, E for *Escherichia coli*, and P representing *P. aeruginosa,* while the remaining *Enterobacteriaceae* were coded with M followed by a number. Re-culturing of bacterial isolates was done by basic medical microbiological techniques. Thus, the bacterial isolates were again plated out on MacConkey agar and cultured aerobically at 37 °C for 24 h. The resulting overnight growth of fresh bacteria colonies was then used for bacteria identification and antimicrobial susceptibility testing using Gram-negative (GN) ID and AST cards of the Vitek 2 Compact Automated System (BioMerieux, Marcy L’Etoile, France), following the guidelines of the manufacturer. Briefly, under sterile conditions, a pure overnight-grown bacterial colony on MacConkey agar was suspended in 3 mL of 0.45% sterile saline solution in a test tube. A turbidity of 0.50–0.63 was obtained using the DensiChek™ (BioMérieux Inc DensiCHECK™) turbidity meter according to the manufacturer’s guidelines. The ID cards were inoculated with the bacterial suspension of each of the bacterial pathogens and placed into the cassette for identification as provided in the guidelines of the manufacturers (https://www.epa.gov/sites/default/files/2017-01/documents/qc-22-04.pdf, accessed on 19 June 2021). The tested antibiotics were ampicillin/sulbactam (AMS), amoxicillin/clavulanic acid (AUG), piperacillin/tazobactam (PTZ), ceftazidime (CAZ), cefepime (PIME), aztreonam (AZT), imipenem (IMP), meropenem (MER), amikacin (AMK), gentamicin (GM), tobramycin (TOB), ciprofloxacin (CIP), levofloxacin (LEVO), Bactrim (BAC), minocycline (MIN), tigecycline (TIG), colistin (CS), trimethoprim/sulfamethoxazole (SXT), ticarcillin/clavulanic acid (TCC), linezolid (LZD), vancomycin (VAN), cefuroxime (CTX), metronidazole (MET), doxycycline (DOX), ceftazidime/avibactam (CAV-AVI), tazobactam (TAZ), and nitrofurantoin (NIT). The minimum inhibitory concentration for the tested antibiotics and ESBL production were also determined using the Vitek 2 Compact Automated System (BioMerieux, Marcy L’Etoile, France).

### 2.4. Definition of Infections

Based on the results of the antimicrobial susceptibility test, the infections were defined according to the international recommendations by the European Centre for Disease Prevention and Control (ECDC) and Centres for Disease Control and Prevention (CDC). Thus, infections were categorised as multidrug-resistant (MDR), extensive drug-resistant (XDR), or pan-drug-resistant (PDR) [12]. In addition, organisms that were resistant to the carbapenems were classified as carbapenem-resistant *Enterobacteriaceae* (CRE).

### 2.5. Antimicrobial Treatment Regime Assay and Determination of Clinical Outcomes

The antimicrobial therapies were categorised either as monotherapy or as combined therapy based on the number of antibiotics used in the patient’s treatment. Clinical outcomes were categorised as either alive and improved (AI) or alive and not improved (ANI), while mortality was documented as indicated in the medical records. The length of hospital stay was taken as the number of days the patient was hospitalised.

### 2.6. Statistical Analysis

GraphPad Prism, version 10.0.2 (232), was used for data analysis. Two-tailed Z-score analysis tests were used to compare the frequency distribution of specimen types with *p* < 0.05 considered as statistically significant. IBM SPSS version 26 was also used to compute statistical differences. Pearson Chi-Square was used to compare the relationship between age, clinical diagnosis, treatment types, antimicrobial resistant pattern, clinical outcomes, and length of hospital stay (LOHS). Significance was taken at *p* < 0.05. The results on antimicrobial susceptibility are presented as percentages, while the RM one-way ANOVA test was used to compare significant differences between mono and combined therapies in the treatment of infections. The length of hospital stays (days) is presented as mean ± SEM (standard error of the mean).

## 3. Results

### 3.1. Demographic Characteristics

The patients included males (45%), females (50%), and 5% whose gender was not specified (NS). Their ages ranged from 3 months to 97 years (Table 1). Most of the patients’ samples (24.7%) were collected in the Intensive Care Unit (ICU) and ER (14.9%). Differences in the number of samples from both wards were not significant (*p* = 0.07). The remaining samples were from 18 other departments of the hospital, which included Cardiology, Ear, Nose and Throat (ENT), Emergency (ER), Family Medicine, General Surgery, Geriatric Medicine, Infectious Diseases, Internal Medicine, Nephrology, Neurology, Neurosurgery, ObGYNE, Oncology, Orthopaedic, Paediatrics, and Urology (Table 1). Of the samples submitted to the laboratory, the most common were urine (38%), followed by wound swabs (15%), and the differences in percentage numbers were significant (*p* = 0.0002), as shown in Table 1

### 3.2. Bacterial Infections and Antimicrobial Assay

Figure 1A displays the number of samples collected and the isolated microorganisms. The results showed that *Acinetobacter baumannii*, *Escherichia coli*, *Klebsiella pneumoniae*, and *Pseudomonas aeruginosa* were mostly isolated from multiple sites. Several other bacterial strains were isolated but were less common in comparison. Figure 1B illustrates the number of samples from patients with urinary tract infections (UTIs) and the causative bacterial agents. *E. coli* was the major pathogen isolated from most samples, followed by *K. pneumoniae* and *P. aeruginosa,* but were detected less frequently as compared to *E. coli*. Figure 1C describes the number of samples with bloodstream infections (BSIs). *E. coli*, *K. pneumoniae*, *P. mirabilis*, and *S. marcescens* were more associated with BSIs as compared to other pathogens. The number of incision drainage (INsD) samples and the pathogens isolated from them are represented in Figure 1D. Only three bacterial pathogens were isolated: *A. baumannii*, *E. coli*, and *K. pneumoniae*. *E. coli* was the most frequently detected microorganism. Figure 1E describes the number of samples and pathogens isolated from sacral wound cultures (SWCs). *A. baumannii* and *K. pneumoniae* were isolated from most SWC samples. Figure 1F displays the number of samples and bacteria isolated from transtracheal aspirate swabs (TTASs). *A. baumannii*, *K. pneumoniae*, *P. aeruginosa*, and *E. cloacae* were isolated, with *A. baumannii* being the most common pathogen. Figure 1G establishes the number of wound samples (WS) and the isolated bacteria pathogens. *A. baumannii*, *E. coli*, *K. pneumoniae*, *P. aeruginosa*, and *Klebsiella aerogenes* were isolated from wound samples. *A. baumannii* appeared to be associated with most wound samples. Figure 1H shows the sputum samples (SP) and their associated pathogens. Only three pathogens were isolated from SP samples. Again, *A. baumannii* was the most frequently detected, followed by *K. pneumoniae* and *P. aeruginosa*. *K. pneumoniae* and *P. aeruginosa* were detected in the same number of samples.

Overall, the pathogens associated with infections in hospitalised patients were *Acinetobacter baumannii* (28%), *Escherichia coli* (26%), *Klebsiella pneumoniae* (22%) as well as other *Klebsiella* species (1%), *Pseudomonas aeruginosa* (12%), *Proteus mirabilis* (4%), *Enterobacter Cloacae* (1%), *Providencia stuartii* (1%), *Shigella flexneri* (1%), and Serratia marcescens (1%) (Figure 1). They varied in the types of specimens from which they were isolated. While *E. coli* was the most common bacteria in urine specimens, *A. baumannii* was encountered in many specimen types except for bloodstream infections. However, *K. pneumoniae* was observed in most of the specimen types (Figure 1A–H).

Figure 2A–D illustrate the antimicrobial resistant patterns and a heatmap of the resistance profile of *A. baumannii*, *E. coli* (A), *K. pneumoniae*, *P. aeruginosa*, and other *Enterobacteriaceae* (B) responsible for hospital-associated infections.

Figure 2A illustrates the antibiotics employed for the treatment of patients with various bacterial infections. The result showed that isolates tested against amoxicillin (AMX), ampicillin (AMP), cefalotin (CF), and colistin (CS) displayed 100% resistance. Furthermore, ticarcillin/clavulanic acid (TCC), levofloxacin (LEVO), tigecycline (TG), and azithromycin (AZT) revealed between 90 and 98% resistance. Amikacin (AMK), cefoxitin (FOX), ceftriaxone (CRO), and gentamicin (GM) displayed resistance patterns of between 30 and 46%, whereas the remaining antibiotics were within a range of 50–87% resistance.

Figure 2B is a heatmap illustrating the levels of resistance displayed by isolates to antibiotics as defined by the ECDC and CDC. Analysis of the heatmap indicates that out of all the isolates sampled from various sources, only five *E. coli* isolates were susceptible (SS) to antibiotics; the rest were either MDR or CRE. However, all *A. baumannii* isolates were either MDR or XDR, with no susceptible isolates.

Figure 2C is a heatmap presenting the resistance levels of *K. pneumoniae*, *P. aeruginosa*, and other *Enterobacteriaceae* isolates to antibiotics. Only seven isolates were susceptible (SS) to antibiotics. *K. pneumoniae* isolates exhibited mostly the CRE type of resistance, whereas *P. aeruginosa* were either MDR or XDR. The rest of the *Enterobacteriaceae* isolates followed a similar pattern, being MDR or CRE with only one susceptible isolate.

Figure 2D is a summary of the overall resistant profiles of the bacterial pathogens.

Overall, 11.9, 40.6, 21.8, and 24.8% of isolates were susceptible to MDR, CRE, and XDR, respectively. XDR = extensive drug resistance, MDR = multidrug-resistant, CRE = carbapenem-resistant *Enterobacteriaceae*, SS = susceptible strain, NS = not specified. The presented data illustrates the comprehensive resistance profiles of all the isolates.

### 3.3. Duration of Hospital Stay, Patients’ Comorbid Conditions, and Clinical Outcomes after Treatment

Table 2 describes the characteristics of the patients with *A. baumannii* infections, including their age, resistance profile, antibiotic treatments, and clinical outcomes. It also displays the different clinical diagnoses associated with this bacterium. Analysis of results showed that the age of the patients was significantly associated with clinical diagnosis and LOHS (*p* < 0.001) using Pearson Chi-Square. At the same time, no association was found with treatment type, antimicrobial resistance pattern and clinical outcome. However, mortality was highest between the ages of 56 and 87 years, while most of the clinical diagnoses were lung-related, representing 21% of all the reported infections. In addition to this, 14% of hospital-associated infections (HAIs) were due to hospitalisation. The results also indicate that 36% of *A. baumannii* isolated from patients with different clinical diagnoses were MDR, while 64% were XDR. Overall, 57% of treatments were combined therapies, and 29% were monotherapies. The mortality rate for patients suffering from *A. baumannii* infections was 25%; however, another 25% of the patients survived but did not improve. Only 36% of patients improved and were subsequently discharged.

Table 3 indicates that the patients infected with *K. pneumoniae* presented with different ages ranging from 3 months old to 88 years old. However, there was no statistical difference between age and clinical diagnosis, treatment, resistance pattern, LOHS, mortality, or no improvement using Pearson Chi-Square analysis. Clinically diagnosed UTIs due to *K. pneumoniae* represented 50% of the overall HAIs attributed to this pathogen. Of these, 37.5% of the patients were treated with monotherapy, while the remaining (62.5%) were given combined therapy. However, there were no significant differences in clinical outcomes in terms of LOHS. Regarding antibiotic treatments, 75% of the isolates from the patients were CRE, 8.3% of the isolates were MDR, and only four isolates (representing 16.7%) were susceptible to carbapenem treatment. The patients’ survival rates also varied, with 33.3% not surviving, 25% alive but not improved, and 41.7% recovered and discharged.

Table 4 displays the ages, diagnoses, treatments, and clinical outcomes of the patients with *E. coli* infections. The characteristics of the infections did not reveal any association with the patient’s age, which ranged from 12 months to 87 years, using Pearson Chi-Square. UTIs with a frequency of 69% were the most common clinical diagnoses; however, we found that 19.2% of other HAIs were due to *E. coli*. Here, antibiotic treatments included both monotherapy (65%) and combined (30.7%) therapy. The response to antibiotic therapy also varied as 19% of the infections were sensitive to antibiotics, whereas 73% of them were MDR, with 7.7% being CRE. In terms of clinical outcomes, the data analysis revealed an 11.5% mortality rate, and 34.6% of patients survived but did not improve. Overall, 53.9% of the patients treated with both monotherapy and combined therapy improved and were discharged.

Table 5 displays the age, clinical diagnosis, antibiotic therapy, resistance patterns, and LOHS of patients with *P. aeruginosa* infections. The ages of the patients ranged from 22 to 93 years; using Pearson Chi-Square analysis, age had no association with infection type or mode of treatment with antibiotics. Most of the clinical diagnoses were UTIs (33.3%), followed by chronic otitis media (25%). The antibiotic therapy for these patients was also both mono and combined therapy, each of which was used to treat 50% of each of these patients. Analysis of *P. aeruginosa* HAI susceptibility to antibiotic monotherapy illustrates that only 16.7% of bacterial pathogens were sensitive, particularly to LEV and GM. However, most isolates of this pathogen were XDR (58.3%), while others were MDR (25%). The patients’ clinical outcomes also varied, with a 33.3% mortality rate, 8.3% of patients alive but did not improve, and 58.4% improved and discharged. In addition, LOHS and clinical diagnoses varied and were mainly due to multidrug resistance.

Table 6 displays the clinical diagnosis, antibiotic treatment, resistance profile, and LOHS of patients infected with *Enterobacteriaceae*. Their ages ranged from 27 to 82 years and showed no statistical difference with clinical diagnosis or LOHS as analysed by Pearson Chi-Square. Although UTIs represented most of the clinical diagnoses with 40%, HAIs were seen in 50% of hospitalisations. *Proteus mirabilis* infections were the most (40%) common clinical diagnosis for this category of patients. Meanwhile, 30% of infections were due to *P. stuartii,* of which the patients were treated with monotherapy, while 40% of the remaining *Enterobacteriaceae* HAIs were treated with combined therapy. The tests for the response to antibiotic therapy showed that only 10% of isolates were susceptible, 70% were MDR, and 20% were identified as CRE. Furthermore, the majority of the patients (70%) survived but did not improve with treatment, while there was a 10% mortality rate. This was also reflected in a longer LOHS.

### 3.4. Monotherapy and Combined Therapy Treatment Types for Clinical Bacterial Infections

The results in Figure 3A–F describe the types of antibiotic treatments used to treat bacterial infections. Figure 3A displays the different HAI bacterial pathogens in this investigation and the mono and combined therapies. Monotherapy was significantly effective in the treatment of *E. coli* infection. In addition to this, the results of either type of treatment for *P. aeruginosa* infections did not show any difference between the number of bacterial infections treated either by mono or combined therapies. Nonetheless, combined therapy appeared to be significantly superior (*p* < 0.05) in the treatment of *A. baumannii* and *K. pneumoniae* infections.

In some patients, combined therapy involved a triple antibiotic regimen. The results presented in Figure 3B describe the different types of combined therapies used for these HAIs. Double antibiotic therapy was found to be significantly (*p* < 0.05) more used than triple antibiotic therapy (Figure 3B).

Figure 3C–F displays what type of treatment pattern produced a better treatment response or no response in terms of either sensitivity or resistance to antibiotics. Our analysis indicated that both mono and combined therapy were used in the treatment of XDR bacterial infections and more for *A. baumannii* infections. In the MDR infections (Figure 3D), mono-therapeutic treatment was used for infections of all bacterial species was observed to be significantly (*p* < 0.05) used for the treatment of MDR *K. pneumoniae*, *E. coli* and other *Enterobacteriaceae* (Figure 3D). At the same time, combined therapy was also used for the treatment of MDR bacterial infections of *A. baumannii* and *P. aeruginosa* (Figure 3D).

Carbapenem-resistant Enterobacteria (CRE) infections were treated with combined therapy, particularly in *K. pneumoniae*, *E. coli*, and *Enterobacteriaceae* infections compared to monotherapy. In terms of response to either monotherapy or combined therapy, treatment for *A. baumannii* did not produce any favourable response. However, *K. pneumoniae* was more sensitive to monotherapy, while *Enterobacteriaceae* showed infections were treated more with a combination of antimicrobials (Figure 3E), while susceptible strains (SS) of *P. aeruginosa* were treated with monotherapy. For the other Enterobacteria, combined therapeutic measures were needed for treatment (Figure 3F).

### 3.5. Comparing the Length of Hospital Stay by Type of Bacterial Infection

Unpaired T-tests comparing the means ± SEM of LOHS between patients with *A. baumannii* and *K. pneumoniae* infections showed no significant difference (*p* = 0.62), meaning that there is no relationship between the duration of hospital stay for both pathogens. However, there were significant differences (*p* = 0.02) in lengths of hospitalisation between patients with *E. coli* infections and those with *A. baumannii* infections, thus indicating a longer hospital stay for patients infected with *A. baumannii* (Figure 4). There were also significant differences between LOHS in patients with *K. pneumoniae* and *E. coli* (*p* = 0.01) infections, while infections of *A. baumannii* and other *Enterobacteriaceae* had no significant differences (*p* = 0.79).

## 4. Discussion

The bacterial pathogens associated with HAIs in this study are similar to those that have been commonly linked to infections in critically ill patients [13,14]. These pathogens *(A. baumannii*, *E. coli*, *K. pneumoniae*, and *P. aeruginosa)* have also been grouped on the list of high global priority pathogens in HAIs [1,15]. In the same context, the highly resistant profiles of the HAIs, as reported here, are also the global norm that has been reported in hospital-associated drug-resistant infections (HARIs) that are currently not treatable with available antibiotics [1,16,17]. The findings in this present report showed variations in the incidence of HAIs by departments and wards (Table 1). However, those from the ICU were more frequent, which is expected since these patients are in critical health conditions with different inserted medical devices [9]. Furthermore, the high patient numbers in ICUs are due to the acceptance of patients from different parts of a region [13]. Thus, the increased incidence numbers of ICU patients associated with HAIs are consistent with those of other reports in the region of the present study [13,18,19] and in other regions of the world [9]. However, with regard to the specimen type, urine and wound samples were the most common in the present study. (Table 1). This is in line with the findings of earlier reports that cited urinary tract infections (UTIs) and surgical wound infections as common HAIs [9,20,21].

With the increased prevalence of HAI-associated pathogens, the findings reported here differ from those of other researchers either in the region of this study or other regions of the world. The high incidence of *A. baumannii*-associated HAIs (28%) seen in this investigation, compared to other bacterial pathogens (*E. coli*, *K. pneumoniae*, *P. aeruginosa*, and other enterobacteria) are contrary to those of other reports [22]. This difference might be due to the genetic strain of the pathogens circulating in different hospitals [23,24], and this could also explain the differences in the HAI *E. coli* pathogens in this investigation compared to those of an earlier report [22]. Also, *E. coli* was reportedly the most common pathogen responsible for HAIs in the European Union (EU) [25], indicating variability between regions.

Therefore, the findings of *A. baumannii*-associated HAIs, which were higher as compared to those of other pathogens in this report (Figure 1A), simply highlight the growing threat of this bacterial species in this region [26]. The widespread nature of this opportunistic pathogen has been reported in hospital settings [27,28]. This bacterium has the ability to cause a wide range of infections in immunocompromised patients [29,30], with outbreaks having been reported in an adult ICU [31]. The resistance profile of the *A. baumannii* HAIs, which were either MDR (36%) or XDR (64%) to tested antibiotics reported here, is worrisome. Overall, the rise of drug-resistant *A. baumannii* continues to gain the attention of researchers around the world [32,33]. However, the pattern of resistance is influenced by factors such as the patient’s susceptibility to the infection due to underlying medical conditions [33]. Hence, it is therefore pertinent to say that the underlying medical conditions and their association with HAI clinical outcomes are poorly quantified. Of the 101 overall HAI cases in this study, 24 (23.76%) died (Table 2, Table 3, Table 4, Table 5 and Table 6). The patient’s age was not a risk factor. Thus, the number of ‘alive, not improved’ cases could be due to comorbid medical conditions and MDR HAI pathogens. HAI resistance profiles (MDR, XDR, and PDR) do not correlate well with the clinical outcomes of patients [13,34]. Although significant differences were seen here between survival (75.25%) and mortality rates (23.76%), this mortality rate was not high when compared to those in other reports [13,35]. There is the possibility that mortality in critically ill patients could be attributed to difficult-to-treat HAIs. Therefore, variations in mortality rates could depend on differences in the management of HADRIs, together with other comorbid conditions.

The incidence of increased antimicrobial resistance of bacterial pathogens has led to limited options for patient management by clinicians, particularly as antibiotics are not being produced as fast as they are needed. On the other hand, the majority of physicians prefer the use of a combination of antimicrobials to that of monotherapy [36]. This practice has both advantages and disadvantages in terms of bacterial resistance. The therapeutic options varied with GNB-HAIs in this study. Monotherapy was preferred for *E. coli* infections (Figure 3A), meaning that positive outcomes were attained by single antibiotic treatment with the antibiotics listed here for this bacterium, which are in line with those used for specific outcomes in *E. coli* infections [37]. However, combined therapy was preferred in treatments of *A. baumannii* and *K. pneumoniae* HAIs (Figure 3A), which could be attributed to the resistant nature of the bacterial pathogens (MDR, XDR CRE). The documented antibiotic treatments for HAIs caused by *A. baumannii* in this study are in line with those of recent recommendations [38], except for some cases where vancomycin was included in therapies. Vancomycin was used either as a monotherapy or in combination with meropenem (Table 2). This inclusion of vancomycin, an antibiotic used for treating MDR Gram-positive bacteria, in treating GNB-HAIs can be attributed to the suggestion that carbapenem-resistant *A. baumannii* are inactivated by vancomycin derivatives [39] in experimental models. However, there are no guidelines or published studies on this. There is a possibility that vancomycin could be used as a prophylaxis to prevent secondary infections in critically ill patients. Most vancomycin therapies in this study resulted in patient outcomes of either alive, not improved or dead, showing that there was no added advantage in the use of vancomycin in the treatment of GNB isolates associated with HAIs. Further investigation is needed to shed more light on this. However, it has been documented by case–control studies that combined and monotherapies displayed no significant differences in observed mortality rates [40]. However, the differences in clinical outcomes, such as mortality between monotherapy and combined therapies, could be determined by the quality of the study [40], with those that are of good quality revealing lower mortality in combined therapy, while those of poor quality indicated no difference.

The result here demonstrated that the clinical outcomes of critically ill patients were governed by a combination of the types of antimicrobial resistance of the HAI GNB pathogens and the existing medical conditions of the patients. There is a need for more investigations that would compare HADRIs in more clinical settings in critically ill patients.

## 5. Conclusions

The current investigation has revealed that there is a high prevalence of XDR/MDR Gram-negative bacterial infections associated with HAIs. *A. baumannii* and *P. aeruginosa*-associated infections produced the highest extensive drug resistance observed in this study. Both mono/combined therapies were used in the treatment of XDR bacterial infections of *A. baumannii* and *P. aeruginosa*, which also had the highest mortality rates. Combined therapy was used for MDR bacterial infections of *A. baumannii* and *P. aeruginosa.* However, treatment for MDR bacterial infections caused by *K. pneumoniae*, *E. coli*, and other *Enterobacteriaceae* were found to respond more to monotherapy. Moreover, the relationship between XDR/MDR Gram-negative bacterial infections correlated significantly with clinical outcomes and treatment. However, no statistically significant difference was observed with LOHS. Overall, this evaluation revealed that although we found an association between age, clinical diagnosis and treatment. However, there was no significant association between age and clinical outcome, resistance pattern, and LOHS. The present study, therefore, highlights the alarming need for strict compliance with hand and routine environmental hygiene in hospital wards to stem the spread of antimicrobial resistance. In addition, continuous surveillance of bacterial species through research of isolates and sharing this information should be encouraged.

## Figures and Tables

**Figure 1 antibiotics-12-01425-f001:**
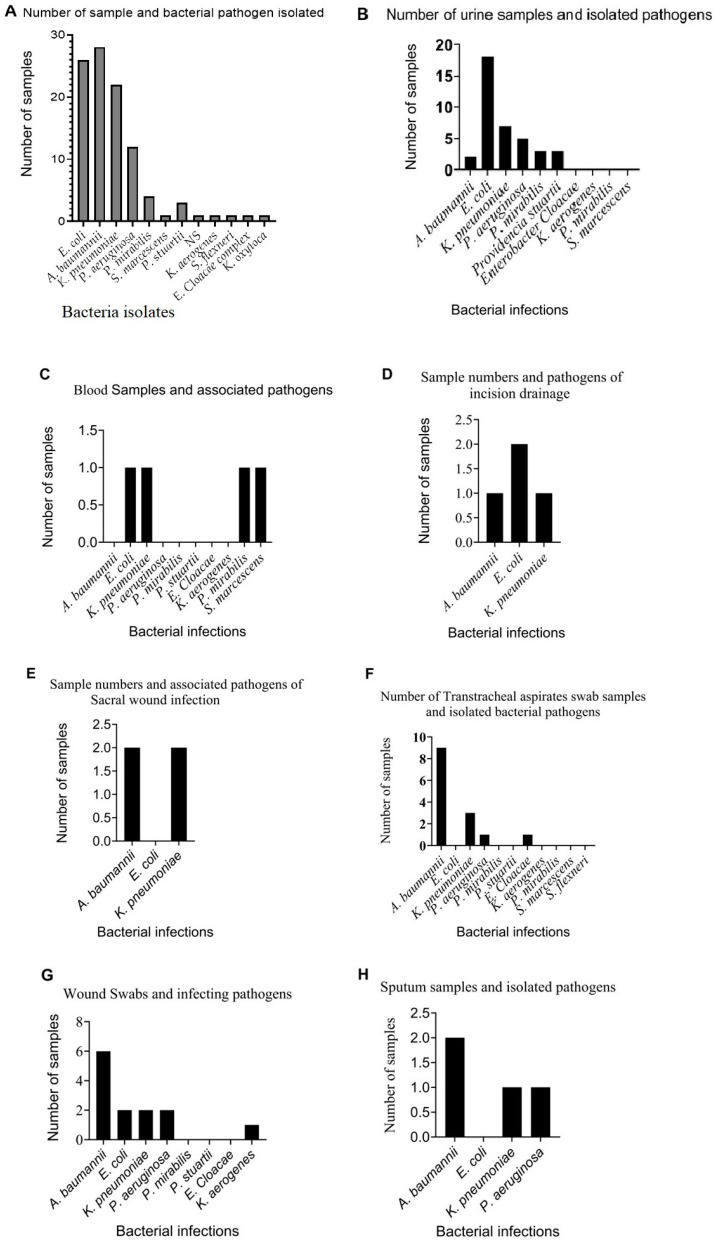
Bacterial pathogens associated with infections in hospitalised patients and their distribution in the clinical samples. (**A**) represents bacterial pathogens from the samples, (**B**) indicates pathogens from urine samples, (**C**) pathogens from blood samples, and (**D**) illustrates pathogens from incision drainage. Those of (**E**–**H**) represent bacterial pathogens from sacral wound infections, transtracheal aspirates, wound swabs, and sputum, respectively.

**Figure 2 antibiotics-12-01425-f002:**
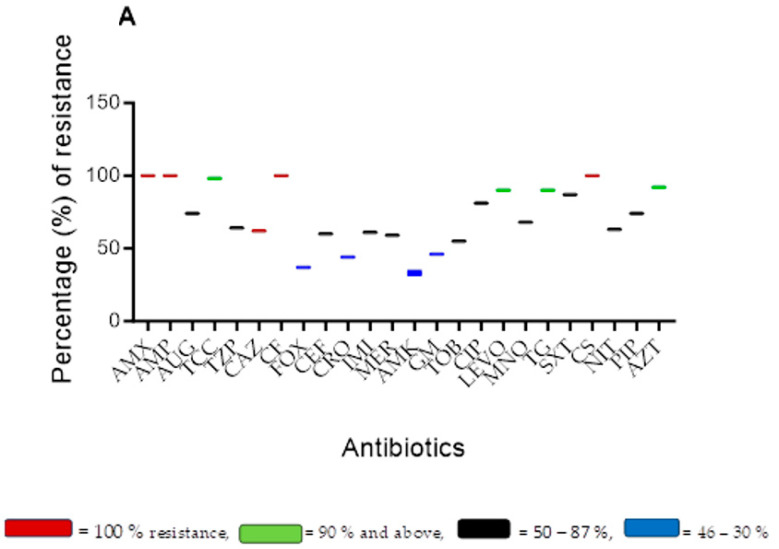
(**A**) Overall antimicrobial resistance pattern of bacterial pathogens associated with HAIs. Heatmap of individual and overall resistant profiles. (**B**) *A. baumannii* and *E. coli* with A representing *A. baumannii* and E representing *E. coli*; (**C**) *K. pneumoniae*, *P. aeruginosa*, and other *Enterobacteriaceae*, with K representing *Klebsiella*, P for *Pseudomonas*, and M representing other *Enterobacteriaceae* bacterial pathogens responsible for hospital-associated infections. (**D**) Overall resistance profile of the pathogens. 1 = sensitive, 2 = intermediate, 3 = resistant, 0 = not tested. Antibiotics: ampicillin (AMP), amoxicillin (AMX), amoxicillin/clavulanic acid (AUG), piperacillin/tazobactam (TZP), cefalotin (CF), cefoxitin (FOX), ceftazidime (CAZ), ceftriaxone (CRO), cefepime (CEF), imipenem (IMI), meropenem (MER), amikacin (AMK), gentamicin (GM), ciprofloxacin (CIP), tigecycline (TG), nitrofurantoin (NIT), trimethoprim/sulfamethoxazole (SXT), colistin (CS), ticarcillin/clavulanic acid (TCC), linezolid (Lzd), vancomycin (VAN), cefuroxime (CTX), metronidazole (MET), doxycycline (DOX), levofloxacin (LEVO), azithromycin (AZT), piperacillin (PIP), minocycline (MNO), nitrofurantoin (NIT), tobramycin (TOB).

**Figure 3 antibiotics-12-01425-f003:**
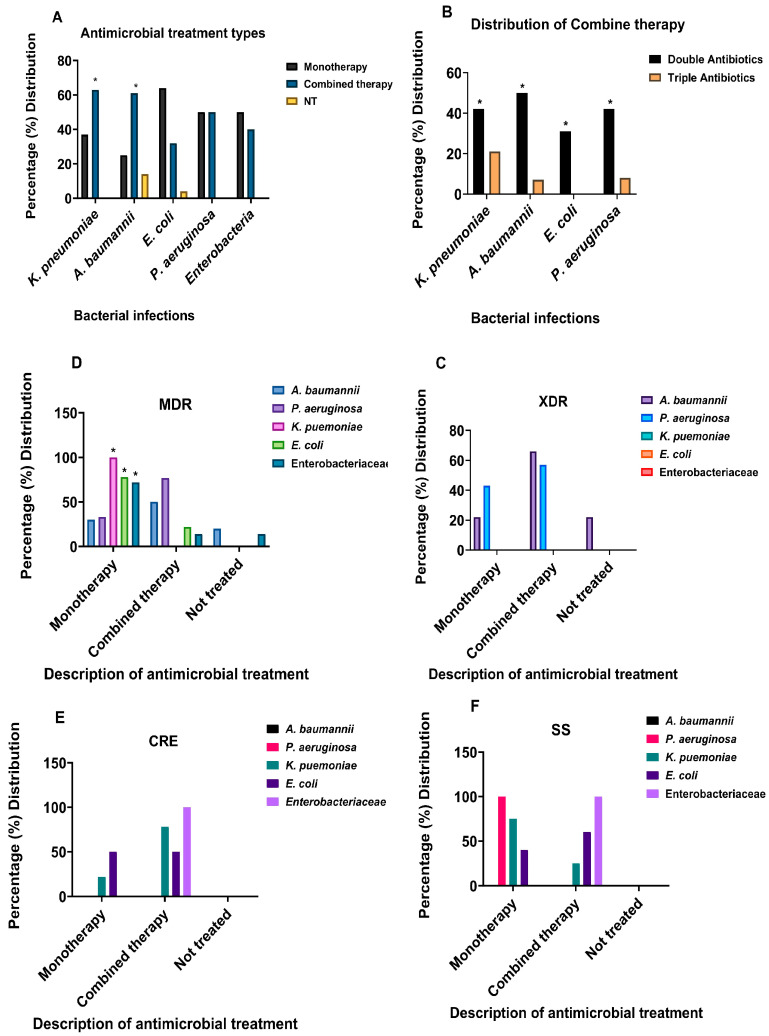
Different bacterial infections with the mode of therapy and resistance profiles. (**A**) Antimicrobial treatment types, monotherapy, and combined therapy for the different HAI bacterial pathogens. (**B**) Types of combined therapies for the different pathogens. (**C**–**F**) Distribution of type of antibiotic treatment for XDR, MDR, CRE, and SS HAI bacterial pathogens. Differences between monotherapy and combined therapy treatments were compared statistically using two-way ANOVA Tukey’s multiple comparison test. * Significance taken at *p* ≤ 0.05. Generally, monotherapy was significantly used in treatments of *K. pneumoniae*, *E. coli* and *Enterobacteriaceae*, while in combined therapies, the combination of two antibiotics was significantly more than those of three. For MDR, a comparison was made between *K. pneumoniae*, *E. coli* and *Enterobacteriaceae,* showing a significant difference with *p* = 0.001. For the remaining pathogen (XDR, CRE and SS), differences between mono and combined therapies were not significant. NT = not treated, XDR = extensive drug-resistant, MDR = multidrug-resistant, CRE = carbapenem-resistant *Enterobacteriaceae*.

**Figure 4 antibiotics-12-01425-f004:**
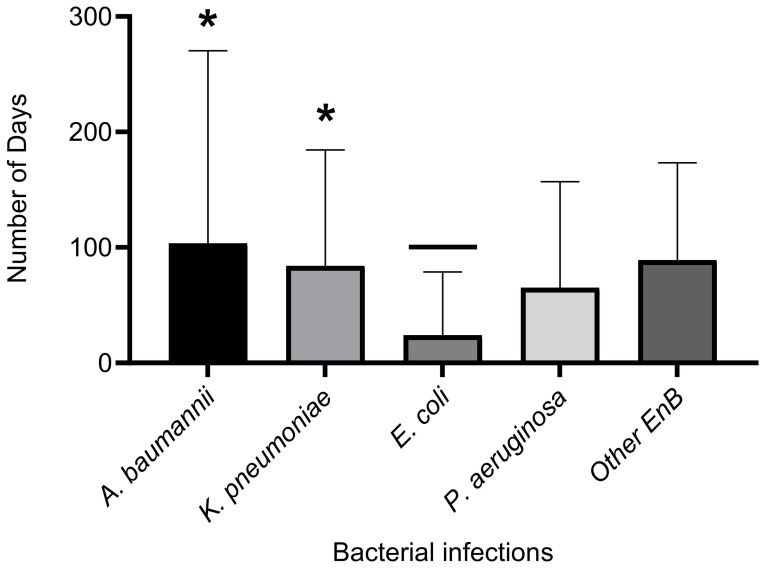
Mean length ± SEM of hospital stay by bacterial infection. Unpaired T-test was used for the comparison of length of hospital stay by bacterial infection with *p* ≤ 0.05 indicating statistical significance. * Indicates significant difference between *E. coli* and *A. baumannii* (*p* = 0.02) and *E. coli* and *K. pneumoniae* infections (*p* = 0.01). EnB = other *Enterobacteriaceae*, SEM = standard error of the mean.

**Table 1 antibiotics-12-01425-t001:** Patient demographics, department of hospitalisation, and type of specimens.

	Age Group	Frequency (N)	Percentage (%)	*p*-Value
Age	0-10 10-20 21-30 31-40 41-50 51-60 61-70 71-80 81-90 91-100	8 1 13 11 10 18 17 10 12 1	7.92 0.99 12.8 10.9 9.9 17.8 16.8 9.9 12 0.99	0.00 ^a^
Gender	Females	51	50	
	Males	45	45	0.91 ^b^
	NS	5	5	
Departments (Hospital wards)	Cardiology ENT ER Family medicine General surgery Geriatric medicine ICU Infectious diseases Internal medicine Nephrology Neurology Neurosurgery NS ObGYNE Oncology Orthopaedic Paediatrics Urology	2 4 15 2 9 2 25 1 10 1 11 4 5 3 2 2 2 1	2 4 14.9 * 2 8.9 2 24.7 * 1 9.9 1 10.9 4 5 3 1.9 1.9 1.9 1	0.07672 ^c^
Type of specimens	Wound swabs Blood Catheter urine Cephalic wound culture Ear swab Gluteal wound Incision Drainage NS Peritoneal fluid Rectal Swab Right maxillary sinus Sputum Tissue Culture Stool Tracheostomy Swab Transtracheal Aspirate Swab Urine	15 4 3 1 3 1 4 5 2 1 1 4 2 1 1 14 38	15 * 4 3 1 3 1 4 5 2 1 1 4 2 1 1 14 38 *	0.00024 ^d^

^a^ Represents statistical significance comparing all age groups, ^b^ signifies statistical comparison for gender. ^c^ is the statistical comparison *p* value for ER and ICU, while ^d^ is the statistical comparison for specimen types. * are the compared values. A significant difference was taken at *p* < 0.05. N = number, NS = not specified, ICU = Intensive Care Unit, ENT = Ear, Nose and Throat, ER = Emergency, ObGYNE = Obstetrics and Gynaecology.

**Table 2 antibiotics-12-01425-t002:** Clinical outcomes of *Acinetobacter baumannii* infections with resistance profiles and therapeutic antibiotic treatments.

Case No.	Age	Clinical Diagnosis	Therapeutic Description (Antibiotics)	Susceptibility Pattern	CLO	LOHS
4	72	Pneumonia, Pulmonary edema	Combined (MER, VAN) *	XDR	Died	3
14	33	Chest and abdomen necrotizing fasciitis	Combined (TAZ, TG)	MDR	AI	18
16	54	Necrotizing fasciitis and Multiple comorbidities	Mono therapy (MER)	MDR	ANI	54
17	NI	NI	NI	XDR	NI	NI
20	79	Tumour	Mono therapy (VAN) *	XDR	Died	46
23	27	Chronic ulcer of skin	Mono therapy (TAZ)	MDR	AI	817
29	24	Pneumonia and Bacteraemia	Combined (TZP, TG)	MDR	AI	52
30	NI	NI	NI	XDR	NI	NI
34	66	Pneumonia, Urosepsis	Combined (TAZ, LEVO)	MDR	Died	135
36	64	Stroke, UTI	Mono therapy (TAZ)	XDR	AI	57
40	56	Benign neoplasm of pituitary gland	Mono therapy (MER)	XDR	Died	30
42	59	Myocardial infraction	Mono therapy (MER)	XDR	Died	49
44	NI	NI	NI	MDR	NI	NI
45	17	Surgical site infection	Combined (VAN, MER) *	XDR	AI	28
46	58	Sepsis, Infected diabetic foot	Mono therapy (AUG)	MDR	AI	54
50	60	Bed sore	Combined (CAV-AVI, VAN) *	XDR	ANI	171
55	58	Intracranial haemorrhage	Combined (CAZ, VAN) *	XDR	Died	13
57	NI	NI	NI	MDR	NI	NI
58	61	Post gastric Sleeve peritonitis	Combined (GM, TG).	XDR	AI	62
66	41	Cephalic wound Infection	Combined (MER, VAN) *	MDR	AI	307
68	85	Complicated UTI	Combined (VAN, CIP) *	XDR	AI	16
72	27	Fever of Unknown origin	Combined (TAZ, TG).	XDR	AI	69
78	63	Bacterial pneumonia, Multiple comorbidities	Combined (CAV-AVI, TG).	XDR	ANI	59
82	87	Acute pulmonary edema	Combined (TG, CS)	XDR	Died	180
83	33	Tracheal perforation	Combined (MER, VAN) *	XDR	ANI	48
88	54	DM II, infective endocarditis	Mono therapy (CIP)	XDR	ANI	17
89	55	Sepsis	Combined (MER, CS)	XDR	ANI	135
91	27	Health care associated meningitis.	Combined (MER, VAN) *	MDR	ANI	65

LOHS = length of hospital stay, CLO = clinical outcome, AI = alive improved, ANI = alive not improved, NI = not indicated. N (total number) = 28. * Represents use of unapproved antibiotic treatment for *A. baumannii*; 25% of patients that received a combination with vancomycin either died or did not improve.

**Table 3 antibiotics-12-01425-t003:** Cases with Klebsiella pneumoniae infections: age, diagnosis, antibiotic treatment, and clinical outcomes.

Case No.	Age	Clinical Diagnosis	Therapeutic Description (Antibiotics)	Resistance Pattern	CLO	LOHS (Days)
1	35	Tumour and HAI Meningitis	Combined (MER, CIP)	CRE	Died	316
2	60	MMA adenocarcinoma	Monotherapy (AUG)	SS	AI	14
5	26	UTI	Monotherapy (CIP)	SS	AI	1
13	56	Nasal polyp	Monotherapy (AUG)	SS	AI	1
15	49	Leg chronic ulcer	Monotherapy (TAZ)	MDR	AI	117
18	78	UTI catheter, Bedsore, comorbidities	Combined (CIP, TAZ)	CRE	AI	3
21	83	Multiple comorbidities	Combined (CLOXA, CAZ-AVI, TAZ)	CRE	Died	7
27	55	UTI	Combined (GM, CAZ-AVI)	CRE	AI	13
31	60	UTI	Combined (CAV-AVI, VAN) *	CRE	ANI	171
39	3 mths	Perforated auns, UTI	Monotherapy (MER)	CRE	AI	14
52	45	UTI	Combined (CEF, VAN) *	CRE	ANI	1
59	82	Septic shock and Pyelonephritis	Combined (CAV-AVI)	CRE	Died	73
64	85	Complicated UTI	Combined (VAN, CIP) *	CRE	Died	16
67	62	Infected Bedsore	Monotherapy (TG)	CRE	Died	279
69	88	Infected Bedsore	Monotherapy (MER)	CRE	Died	16
70	61	Pneumonia	Combined (CIP, TG)	CRE	AI	14
76	63	Bacterial pneumonia, UTI	Combined (CAV-AVI, TG)	CRE	ANI	59
85	40	UTI	Combined (AUG, AMP)	SS	AI	1
86	77	UTI	Combined (TAZ, GM)	CRE	ANI	264
87	41	UTI	Monotherapy (CIP)	MDR	AI	1
93	55	Sepsis	Combined (MER, CS)	CRE	ANI	135
95	64	Spine Infection, and Brucella	Combined (CIP, DOX)	CRE	ANI	173
96	12 mths	Bacterial meningitis	Monotherapy (MER)	CRE	Died	167
97	60	Urosepsis and Aspiration pneumonia	Combined (CAV-AVI, CS, GM)	CRE	Died	159

LOHS = length of hospital stay, CLO = clinical outcomes, mths = months, AI = alive improve, ANI = alive not improved. * Represents use of unapproved antibiotic treatment for *K. pneumoniae*. All patients who received a combination with vancomycin either did not improve or died.

**Table 4 antibiotics-12-01425-t004:** Cases with *E. coli* infections: age, diagnosis, antibiotic treatment, and clinical outcomes.

Case No.	Age	Infection and Clinical Diagnosis	Therapeutic Description (Antibiotics)	Antibiotics Profile	CLO	LOHS
3	47	VZS and secondary Bacterial infection	Combined (CIP, LZD)	MDR	AI	18
7	34	UTI	Monotherapy (CIP)	SS	AI	1
8	64	UTI	Combined (CIP, T/S)	SS	AI	1
11	44	UTI	Combined (AUG, T/S)	SS	AI	1
19	72	UTI and Multiple comorbidities	Combined (BAC, CIP)	MDR	AI	1
22	49	Acute cystitis with Multiple comorbidities	Monotherapy (AUG)	MDR	AI	1
26	27	Acute cystitis	Monotherapy (NIT)	MDR	AI	1
28	80	Stage 4 Bed sore	Monotherapy (CEF)	MDR	Died	80
32	39	Sepsis and bilateral thigh abscess	Combined (VAN. TAZ) *	SS	AI	21
35	72	UTI	Monotherapy (NIT)	MDR	AI	1
37	64	Left PCA stroke, UTI	Monotherapy (TAZ)	MDR	AI	57
38	87	Infected Bed sore and UTI	Combined (CRO, CAZ)	MDR	Died	31
41	24	Ileocecal stricture	Monotherapy (AUG)	MDR	AI	30
47	36	UTI	Monotherapy (CTX)	SS	AI	1
48	37	UTI	Monotherapy (CRO)	MDR	AI	1
49	82	UTI	Monotherapy (TAZ)	MDR	ANI	1
54	90	Complicated UTI	Combined (AUG, NIT)	MDR	ANI	1
61	23	UTI	No antibiotic given	MDR	ANI	1
71	31	Hydronephrosis, Sepsis	Monotherapy (CIP)	MDR	AI	17
74	57	Viral pneumonia and UTI	Monotherapy (TAZ)	MDR	ANI	30
77	12 mths	UTI	Monotherapy (AUG)	MDR	ANI	1
79	65	Post gastric bypass leak	Combined (IMI, VAN) *	CRE	ANI	60
94	29	UTI (case of Sickle Cell Disease)	Monotherapy (T/S)	MDR	ANI	1
98	72	UTI	Monotherapy (CIP)	MDR	ANI	1
99	54	UTI	Monotherapy (AUG)	MDR	ANI	2
100	77	UTI	Monotherapy (CS)	CRE	Died	264

LOHS = length of hospital stay, mths = months, CLO = clinical outcome, AI = alive improved, ANI = alive not improved. * Represents the use of unapproved antibiotics for *E. coli*.

**Table 5 antibiotics-12-01425-t005:** HAIs of *Pseudomonas aeruginosa*, associated medical conditions, antibiotic therapy, and clinical outcomes.

Case No.	Age	Infection and Clinical Diagnosis	Therapeutic Description (Antibiotics)	Susceptibility Profile	CLO	LOHS (Days)
6	40	UTI with Vaso occlusive crisis	Monotherapy (LEVO)	SS	AI	1
12	22	Bilateral otitis externa	Monotherapy (GM)	SS	AI	1
24	44	Nasal Polyp, Chronic otitis media	Combine (TobraDex, AUG)	XDR	AI	1
25	93	Right MCA stroke	Combine (CRO, CAZ)	XDR	AI	NI
33	25	Bacterial meningitis	Monotherapy (MER)	MDR	Died	166
51	89	Pneumonia	Monotherapy (TAZ)	XDR	Died	139
60	87	Urosepsis	Monotherapy (GM)	XDR	AI	7
65	69	Leg cellulitis, UTI	Combine (IMI, CIP)	MDR	AI	46
73	61	Sepsis	Combine (GM, TG)	XDR	AI	62
75	62	UTI	Combine (AUG, TG)	MDR	Died	279
84	61	Diabetic foot infection with gangrene	Combine (MER, VAN, TG) *	XDR	Died	14
90	45	Chronic suppurative otitis media	Monotherapy (LEVO)	XDR	ANI	1

* Represents the use of unapproved antibiotics for *Pseudomonas aeruginosa*. Only patients who received a combination with vancomycin died. LOHS = length of hospital stay, CLO = clinical outcome, AI = alive improved, ANI = alive not improved, NI = not indicated.

**Table 6 antibiotics-12-01425-t006:** HAIs of *Enterobacteriaceae*, associated medical conditions, antibiotic therapy, and clinical outcomes.

Case No.	Age	Clinical Diagnosis	Antibiotic Treatment	Infecting Bacteria	Antibiotics Profile	CLO	LOHS (Days)
9	45	UTI, multiple comorbidities	Monotherapy (CEF)	*Proteus mirabilis*	MDR	ANI	1
10	58	Stroke	Combined (VAN, CEF) *	*Enterobacter Cloacae*	SS	ANI	236
43	35	Bacterial meningitis	Monotherapy (BAC)	*Providencia stuartii*	MDR	AI	NI
53	80	Stroke (Bed ridden)	No antibiotic given	*Providencia stuartii*	MDR	Died	202
56	27	Bloody Diarrhoea	Monotherapy (MET)	*Shigella flexneri*	MDR	ANI	1
63	60	Urosepsis and aspiration Pneumonia	Combined (CAV-AVI, CS, GM)	*Proteus mirabilis*	MDR	Died	159
80	63	Bacterial pneumonia, Multiple comorbidities	Combined (CAV-AVI, TG)	*Serratia marcescens*	CRE	ANI	59
81	82	UTI, sepsis	Combined (CAV-AVI, VAN) *	*Proteus mirabilis*	CRE	ANI	73
92	27	HA-pneumonia/ventilatory associated pneumonia.	Monotherapy (TAZ)	*Providencia stuartii*	MDR	ANI	69
101	65	UTI	Monotherapy (SXT)	*Proteus mirabilis*	MDR	AI	1

* Represents the use of unapproved antibiotics for the treatment of *Enterobacteriaceae*. All patients who received a combination with vancomycin did not improve. LOHS = length of hospital stay, CLO = clinical outcome, AI = life improvement, ANI = alive not improved, NT = no treatment.

## Data Availability

Data are available upon request from the authors.

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
