# Peer review of "Incidence of Drug-Resistant Hospital-Associated Gram-Negative Bacterial Infections, the Accompanying Risk Factors, and Clinical Outcomes with Treatment"

_antibiotics, 2023, doi:10.3390/antibiotics12091425_

Round 1

Reviewer 1 Report

Manuscript titled “Incidence of Drug resistant hospital associated Gram-negative bacterial infections, the accompanying risk factors, and clinical 3 outcomes with treatment” have the aim of investigating clinical outcomes with mono and combination therapy of extensively drug-resistant and multi-drug resistant hospital acquired infections.

Please read the author’s guidelines for correct formatting, abstract contents of the article.

No consistency of the information in the affiliations: for example some street address is given in some but not in others but country is provide in some and not in others. It is recommended that the untidiness also of the affiliation be corrected as the alignment is off on some and in line 8 there is m there that should not be.

There are far too many language issues in the text to appreciate the real understanding of this study or impact. Language is difficult and makes it hard to understand the statements.

*For example “Extensive drug resistance in hospitalised patients is accompanied by high morbidity and 16 mortality rates due to limited treatment options.Extensive drug resistance of what??? Bacteria, viruses, protazoa or what?

Abstract uses to many jargon / acronyms that make it difficult to understand what the authors are trying to state. They should be spelled out first and then used.

*For example “This study investigates clinical outcomes with 17 mono and combination therapy in the face of XDR and MDR HAIs.”   What is XDR and MDR HAIs? 

*Similarly in the next sentence “Selected for the study were 18 Cases of HADRIs for hospital wards.” What is meant by 18 Cases of HADRIs and why is Cases capitalized?

*Similarly in the next sentence “Bacteria IDs and AST were ascertained with Vitek 2 Compact 19 Automated System.” What is meant by IDs and AST and why this system used here?

*And the next sentence is not understandable at all with grammar and confusing words used  “Patient treatment types and clinical outcomes (alive improved (AI), alive not 20 improved (ANI), or died while length of hospital stay (LOHS) were instituted from hospital rec-21 ords.”  What is meant by treatment types, (alive improved (AI) use of ( is wrong here, and what is meant by “instituted from ….”.

*Again here too with the acronyms making it hard for the reader to understand if they are not initially spelled out “They were MDR (40.59 %), XDR (24.75 %) and CRE 25 (21.78 %).” And who are they?

*Difficult sentence. Please reword “Treatments were mono/combined therapies with variabilities.” What are the treatments and what are the variabilities? Need to define what is meant by mono therapy and then combined

*Too vague statements without explicit information in it “Clinical outcomes and LOHS varied with infecting 29 bacteria. Prevalence of XDR and MDR HAIs were significantly high with no association to treat-30 ment type, LOHS and outcome.”

*Repetition of words like “generally”,

Inconsistency of wording or grammar:

For example, MDR-GNB,  why is there a hyphen here and not in MDR HAIs, etc

Why is Gram capital in line 42?

*Number association is difficult in this sentence “While it is esti-47 mated that there are 700,000 yearly recorded deaths due to these superbug bacterial in-48 fections globally, this number is postulated to rise to 10 million by the year 2050”. Do you mean 10 M annually or accumulative?  Remove superbug as it is redundant as you have bacterial there and it is colloquial language.

*Joint these two sentences as the second one is convoluted and vague “ Many risk factors are associated 52 with HAIs. One of which is the ward of hospitalization, surgery, and its type as well as 53 the underlying patient’s medical conditions [9].”

*Remove On the other hand as it is inappropriate here, in “On the other hand, with treatment not being delivered as fast as they are required, 55 treatment approaches have evolved that include combinations and alternative options all 56 of which have not led to easy nor successful treatments for critically ill patients [7].”

*Remove However as it is inappropriate here, in “However, MDR-GNB infections are not abating, and difficulties faced by clinicians has 59 led to treatment approaches to include the use of combined therapeutic measure amongst 60 other options”

*Difficult language in the next sentence “Conversely literature on the management of such infections either as sin-61 gle or with combined antimicrobial therapies remain scanty [6].”  Please reword the sentence and use a synonym for scanty as it is colloquial.

*Difficult language “This is with a view to providing more insight into the 65 current therapeutic options in the management of these infections in hospitalised patients 66 from different hospital wards.”  What is meant by “This is with a view”

*Difficult language “The susceptibility to antimicrobials is ascertained, as well 67 as the therapeutic options used in the treatment of the patients instituted” Please reword as it is no clear what you mean by The susceptibility nor therapeutic options.

*Repetition of words “This is aimed at 68 providing gainful insights into the patient outcomes with treatment of difficult to treat 69 bacterial infections in the region of study.”

There is far too many English language issues in the manuscript for the reviewer to comment on and it is not the job of the review to pin point all the language issues. It is advised that this article be proofread by and native English speaker to correct the numerous pitfalls.

Methods

2.2. Study design and data collection

*How can this be a retrospective study when samples are taken from the patients? Do you mean that the samples were from a bio-bank, or?  And was there patient consent in this retrospective study?

* infections of drug resistant bacterial isolates. How was classification of drug resistant bacteria made?

*What is meant by samples in the following statement “A retrospective study involving infections of drug resistant bacterial isolates from 81 hospitalized patients was carried out with samples obtained from the following hospital 82 wards: Cardiology, ENT, ER, Family medicine, General surgery, ICU, infectious diseases, 83 Internal medicine, nephrology, Neurology, Neurosurgery, obstetrics and gynaecology, 84 Oncology, orthopaedics, Paediatrics and Urology.”

*You must elaborate on what lab results exactly was used to provide that these patients had antimicrobial susceptibility of GNB isolates.

*lines 88-90. Again you have to provide explicit information on what was done. Cases means patients, isolates of GNB were in one patient or were there many GNBs in a patient?  So how many patients had GNB in the study, 101 or? Also what is IDs and AST?  These colloquial terms need to be spelled out.

*This sentence does not make sense, Reword it. “Inclusive criteria were that isolates had to be multidrug (MDR) resistant, from 91 hospitalised patients irrespective of age, gender, and nationality.”  Inclusion criteria for the patients or for the isolate as this is a retrospective study on patients that have drug resistant bacteria, Right?

* Excluded were 92 infections of Gram positive bacteria as well as those from outpatient department. You are excluding patients with infection of gram positive bacteria, Right? Why exclude gram positive bacteria and outpatient department?  Needs reasoning here.

*What do you mean by “Based on this initial selection”. It is confusing here.

2.3. Identification and antimicrobial susceptibility assay of bacteria isolates.

*How were these bacteria isolates obtained, biopsy, blood, saliva, etc and how many from each patient and what isolate from each patient?

*More explicit information on the following is needed by thei Vitek 2 Compact System “bacteria identification and antimicrobial 102 susceptibility testing using Gram-negative (GN) ID and AST cards of Vitek 2 Compact 103 Automated System (BioMerieux, Marcy L’Etoile, France), following the guidelines of the 104 manufacturer. “ Provide a short synopsis as not everyone has this system.

*Again more details of how this system does this MIC: Minimum inhibitory concentration for the test 110 antibiotics and ESBL production were determined by Vitek 2 Compact Automated 111 System (BioMerieux, Marcy L’Etoile, France). Provide a short synopsis as not everyone has this system.

2.4. Defining of Infections 114

Based on the results of antimicrobial susceptibility assay, infections were defined 115 according to the international recommendations by European Centre for Disease 116 Prevention and Control (ECDC) and Center for Communicable Diseases (CDC).

*This is confusing as the whole premise was to have samples from patients with drug resistant bacterial isolates. So the patients already were defines of having an infections. Or what is the really the point here? 

2.5. Antimicrobial treatment regime assay and determination of clinical outcomes.

*The following needs more explicit informatin regards what antibiotic or drugs were used jin the mono- or combined therapy. An example is appreciated here? monotherapy or as combined therapy 123 based on the number of antibiotics used in patient’s treatment.

*Difficult wording here too. “Clinical outcomes were 124 categorized as either alive improved (AI) or, alive not improved (ANI).”  Please reword for example “as the patient was alive with improvement, ….”

Spacing between 2.5 and 2.6 is not correct.

Results

*24.7% and 14.9% is not majority.  Please Correct. “Majority (24.7 %) of patient’s samples were from intensive care unit (ICU), and ER (14.9 140 %) besides, differences between samples from both wards were not significant, with 141 p-value 0.7.”

Table 1

What is meant by frequency?  Do you mean Number (N= )?

Why do you have the table title on top and bottom of the table?

What do you mean here of p value? It has to be explicit in the table or its legend or footnotes. And why have it at all in the table as it is not in every row?  It is confusing.

Not all the samples are defined (what is NS), and there is no consistency with the use of capital letters or not. Please tidy these items.

*Confusing statement here. “Of the samples submitted to the laboratory, urine was 38 % and 146 wound swab, 15 % and differences were significant (p-value 0.0002) as shown in Table 1” The comma is out of place and difference were significant from what to what to be significant??? What is meant by the % here, that there 38% of the samples from the infected patients were urine?

*Formatting of the following is incorrect in the manuscript, please correct, or is the Table legend? Or is it supposed to be in the body of the text? “Two-tailed Z-Score analysis was used to compare frequency distribution of specimen types with significance taken at 150 0.05. NS = Not specified, ICU = Intensive care unit, ENT = Ear nose and throat, ER = Emergency, ObGYNE = Obstetrics 151 and gynaecology. P-value significant at p < .05.

*Inconsistent use of capital F or not in figures (in the body of the text) as well as the Table. Please read the author’s guidance on how this is supposed to be.

*More explicit and details comments are needed on every panel in every Figure.  Without explicit description of the figure, the reader will not be able to appreciate fully the results or conclusions made on them. Also Figures as well as Tables need to be places as soon as it is mentioned in the body of the text to make it easier for the reader.

For example: the following for Figure 1 is too vague and does not indicate where the reader is supposed to see the data in Figure 1 as there are 8 panels in Figure 1. It is advised that authors label the panels and provide explicit description for each panel in the text. “The pathogens associated with infections in hospitalised patients were Acinetobacter 155 baumannii (28 %), Escherichia coli (26 %), Klebsiella pneumoniae (22 %) as well as other 156 Klebsiella species, Pseudomonas aeruginosa, Proteus mirabilis and other Enterobacteriaceae 157 (Figure 1). They varied in specimens from which they were isolated. While E. coli was 158 highest in urine specimens, A. baumannii was encountered in a high number of specimen 159 types except for blood stream infections (Figure 1). However, in the majority is K. pneu-160 moniae observed in all the specimen types as shown in the figure 1.”

*What is meant by sample site collection in figure 1? This is confusing as it is not mentioned in the text. Also are these drug resistant pathogens?

*Similarly for Figure 2, more explicit description is needed. For example “In Figure 2, the resistance profile exhibited by the clinical isolates against the twenty-four 162 (24) tested antibiotics is illustrated. Resistance ranged from 33 % (amikacin) to 100 % for 163 amoxicillin, ampicillin, cefoxitin and colistin in cases where these drugs were tested.”  There are four panels in Figure 2. Authors need to explicitly describe all of the panels in dept as the above doesn’t do this. What is meant by “resistance profile exhibited” What is meant by “resistance ranged from”? and why were not all drugs tested on all isolates?  How many isolates were tested against exactly which drug? Figure is far too confusing without more details and explanation of what we are seeing.

The legend is all to confusing as well. Need to make it more clearer.

*Similar comments of the rest of the results. They are poorly described and need better explicit description in the body of the text to understand the authors.

For example, why does Table 2 describe only characteristics of patients who had A. baumannii infections with 200 their ages, resistance profile, antibiotic treatments, and clinical outcomes?   

Then table 3, patients infected with K. pneumoniae only

Then table 4 Table 4 displays patients’ infection profile of E. coli infections

Table 5. HAIs of Pseudomonas aeruginosa,

Table 6. HAIs of Pseudomonas aeruginosa.

Table 5 and 6 have the same title.

All the tables from 2-6 need tidying up. There are acronyms not spelled out in the legend, there are capitals and not capitals, not all column parameters are defined in the text or in the legend.  For example what is meant by resistance profile and then Suceptibility pattern. Then there is statements like “25% of patients that received a combination with vancomycin either died or did not improve” How many patients were treated with vancomycin as percentage means nothing without the total number.

*Authors are also instructed to take note on the use of incorrect tense of the results. Please correct everywhere.

Figure 3 is the best described result in the manuscript but need still improvements with its language. For example the title “Showing different bacterial infections with types of therapy and resistance outcome.” Is to colloquial and what is meant by showing? The tense of the title, etc.

3.5

Unpaired T-Test comparing the means ± SEM of LOHS between A baumannii and K. 375 pneumoniae showed no significant difference (p-value 0.62). This statement needs to be put into context as it means nothing at the moment.

Figure 5 title needs improvement.

Discussion

Authors are encouraged to make explicit language checks and refer back to their specific work when comparing with others.

References need tidying up and some references are not in the correct format.

needs improvement.

Author Response

Comments are attached as pdf

Reviewer 2 Report

And a great research article. Hospital-acquired infections are one cause of sepsis and failure to care for the patient along with sometimes comorbidities. A targeted antibiotic therapy, early recognition of the origin of the infection could be life-saving and/or avoid sequelae. There are some points that need to be clarified to improve reading:

-It is useful to report the total number of patients who are selected and who are subsequently excluded (e.g.; from 1000 excluded 900 included 100).

-The research considered are MDR bacteria. But on tables 3-6 as well as MDR there are also comments of SS, CRE,XDR. This needs to be clarified better.

-Was the research based only on protocols regardless of MDR and pharmacokinetics and route of elimination? Were there different choices for patients with impaired renal function, for example? This needs to be clarified.

-Finally, a summary table is needed with all the pathogens listed with the choices of antibiotic therapy, the methods of administration (for example oral), the days of therapy.

Author Response

Kindly find the attached

Reviewer 3 Report

The manuscript is a retrospective study of 101 gram-negative bacterial infections that were predominantly drug-resistant (~87%) in hospitalized patients at King Fahd Hospital of the University, Al Khobar, Saudi Arabia.   The samples were from a diverse source, including from patients of age groups of 3 months to 93 years old, with multiple types of specimens from urine to incision drainage, and from various hospital wards / departments, including ER and family medicine.  The patients were given mono or combined therapy and the clinical outcomes (CLO) along with length of hospital stay (LOHS) are discussed.  This is a very valuable report providing details for each infection, clinical diagnosis, drug resistance (or susceptibility) pattern and clinical outcome.  However, the manuscript lacks keys information, especially in terms of sample selection, conclusions on the clinical outcomes of treatments (Figure 3 and Tables 2 to 6), clarity in presentation of certain data and written content.  Specific comments to explain the same are included below:

1.    Abstract has valuable information in parts but requires improvement for clarity.

a.    Please expand the various short forms – XDR, MDR, HAI, HADRI.

b.    Lines 17 and 18:  In the current format, clinical outcomes alone of treatments and infections appear not to be the main investigations of the manuscript, with several other important learnings, including type of infections.  Also, discussed further later.

c.    Lines 22 and 23:  Mention of excel and GraphPad Prism is not necessary in the abstract.

d.    Lines 25 and 26:  Please highlight ~12% of susceptible strains (SS) as well.

e.    Line 26:  Unclear what authors mean by “variabilities”.

f.     Lines 26 and 29:  How these conclusions were drawn from Figure 3 are unclear as detailed data in Tables 2 to 6 appear to contradict.  For more details refer to comments on Figure 3 (and Tables 2 to 6).

2.    Lines 86 to 93:  These lines state that patients’ samples were selected based on presence of multidrug resistant isolates, however, 11/101 samples were susceptible strains (SS).  How and why were the SS chosen.  Interestingly, none of the A. baumannii isolates are SS. 

3.    Line 129:  Unclear what “(283)” refers to.

4.    Line 139:  It will be more impactful to mention actual ages, which I believe is 3 months to 97 years old.

5.    Lines 141 to 142 and Table 1:  It is not clear what is being compared (written states ICU is compared to ER but the table suggests frequency of all departments are compared) for statistical significance, and what is the relevance of this comparison for the study.

6.    Lines 146 to 147 and Table 1:  Similar questions as the previous comment.

7.    Table 1 (lines 148 to 152):  Only the headings and last column have margins, but not the rest.  Similar concern with all tables.  Also see above comments #5 and 6 on relevance of p-value and what is being compared is unclear.

8.    Lines 156 to 158:  Please highlight % of all bacteria mentioned as done in the abstract.

9.    Figure 1 (lines 184 to 190): 

a.    Please provide a brief description for figure beyond just a title.

b.    For the first graph on the left-hand side of the page:

                                          i.    Y-axis title is unclear.  I assumed it is just “Number of samples”, and not any specific site of collection.

                                        ii.    Like above, title above the graph is also unclear.

                                       iii.    Species name should be starting with a capitalized letter, please confirm throughout the manuscript as well.

                                       iv.    X-axis:  Kindly be consistent throughout the manuscript about a legend for the bacterial infections – either have one or not, and if have one keep the same name (“Bacterial infections” read better).  Also, same comment applies to Figures 3 and 4.

                                        v.    Suggest splitting this into a subsection of Figure 1A, and specific site samples as Figure 1B.

c.    For specific sample site infection graphs it is unclear why each graph has different bacteria plotted, some only highlight ones detected (like INsD) and others show even ones not detected (all others).

110.  Figure 2:

a.    Please label bacterial infection initials in the figure legend.  Were the “Isolate Lab codes” intended to provide a short form for the respective bacterial infections (A for A. baumannii)?  If so, the ones at the bottom of 2A are not labeled.

b.    Provide a sub-heading for the tables on the left-hand side.

111.   Lines 209 to 212 (and Table 2):  Based on these numbers it is unclear which therapy was advantageous even though the authors conclude combined therapy was significant for clinical outcome.  Furthermore, only about one-half of the treatment was with monotherapy making it difficult to compare and conclude between the two treatment regimens.

112.  Lines 215 to 216:  Statement is unclear.

113.  Line 235:  Please highlight the bacterial infection being referred to.

114.  Table 6, line 305:  Bacterial infection mentioned is incorrect.

115.  Figure 3 and lines 312 to 372:

a.    Please clarify what is the Y-axis measuring. 

b.    It is unclear how does the data for treatment regimens correlate with clinical outcomes.  For example, based on lines 314 and 315 (and Table 4), a higher percent of patients were treated with monotherapy instead of combined therapy to treat E. coli infections, but this alone does not mean that the clinical outcomes (CLO) and/or LOHS were better for monotherapy compared to combined therapy (as shown in Table 4).  Hence, lines 315 and 316 cannot be concluded.  The same applies to the conclusions for other bacterial infections highlighted in this section as well.

c.    Please demonstrate in the graph which values are compared for statistical analysis.

d.    Label fonts are of different sizes and the sub-labels A to F are not aligned with the graphs.

116.  Figure 4:  Helpful figure!

a.    Kindly add lines in the graph to show the bacterial infections being compared for statistical significance (instead of only adding a star above the bacterial infection).

b.    Please clarify Y-axis label to length of hospital stay.

c.    Specify data are mean and SEM and the type of statistical test done in the figure legend as well.

d.    Though not necessary given the nature of the dataset, but it might be more significant to look at statistical analysis by ANNOVA in GraphPad Prism to compare the various infections.

117.  Conclusion, lines 476 to 480:  It is unclear how these conclusions were derived from the data.  Please refer to comments for Figure 3 and Tables 2 to 6 for more details.

11.    Line 77:  “(IRB)” should be before “of”.

22.    Line 83 to 85:  Please be consistent.  For example, “Cardiology” starts with a capital “C”, but “nephrology” starts with small “n”.

33.    Above point applies to Table 1 for “Departments” and “Type of Specimens”.

44.    Line 98:  Mentions number of patient-days in hospital, which I believe is the same as LOHS.  Similar wording has been interchangeably used throughout the manuscript, kindly make sure it is consistent.

55.    Line 105 to 112:  Short forms of antibiotics are not all capitalized unlike in rest of the manuscript.

66.    Line 108:  Short forms are sometimes different, please be consistent.  For example, Gentamicin is referred as Gn here whereas it is GM in other locations.

77.    Lines 162 and 163:  It is not needed to spell out the number 24 and again mention it backets “(24)”.

88.    Lines 163:  Remove brackets for amikacin and add the word “for” before it for consistency.

99.    Line 192 and 193:  For Heatmap and Hospital first letters should be capitalized.  Similar related errors present throughout the manuscript, please correct.

110.  Line 212:  Add “were” before “subsequently”.  

111.  Lines 212:  Additional space before “discharged”, such errors present at several spots throughout the manuscript.

112.  Line 219:  Remove “off”.

113.  All Tables:  Similar to Table 1, please be consistent with the starting with a capitalized letter or not.

114.  Line 408: Should be “another report” or “other reports”.

115.  Line 417:  Missing a word, for example, could add “nature” after “the wide spread” to clarify.

116.  Line 419:  Add “individuals” or “patients” after immunocompromised.

117.  Line 420:  Unclear consider re-wording.  For example, “The resistant profile of A. baumannii HAIs to tested antibiotics reported here is worrisome too.”

118.  Lines 426 to 428 are unclear.

119.  Lines 407 to 411 are unclear, why is “vital discovery” in brackets.

Author Response

Kindly find the attached

Round 2

Reviewer 1 Report

Review 2

In the process of reviewing, the authors are encouraged to provide a thorough and edited version of the manuscript when it comes to formatting and English language issues. The manuscript must be in a format that allow credible review, and this is not the case again in the second version of the manuscript. In addition, it is encouraged that authors respond and correct all the issues the reviewers point out in the first review, as this was not the done here. Hence there are still major issues that need to be addressed in this second version. Please diligently address all the issues and read the author’s guidelines as having 6 pages of issues with the manuscript at this stage of the review process is not advisable. It is fair to say that there are way too many issues with this manuscript to warranty publishing.

With all due respect, this manuscript is untidy and still has formatting and language issues that must be addressed.

For example:

*Alignment in affiliation

*Spacing and manuscript placement for Table 1 and it’s footnote

*Figure 1 has no legend only a title. Spacing of panels within the figure is untidy (not align on top or bottom, text not centered, etc) and  the Figure panels can easily be formatted in portrait not in landscape.

*Figure 2’s footnote is not spaced correctly

*Why is figure 1 title bolded and Figure 2 not?

*Table 2 Title is on its own page. Why?

*Table 4 Title is on its own as well. Why?

*Table 4 is bolded. Why?

*Spacing of Tables footnotes are not consistent. Why?

*Table 5 Title is bolded. Why?

*Table 6 Title is bolded. Why?

*Figure 3 is untidy with no alignment of panels and text. There title is bolded. Why? There is not enough description for the figure to stand on its own. Remove (A-F) in the following as it is not needed “Figure3 (A-F).

*Figure 4 title and the second sentence are bolded. Why? Difficult language that needs rewording “Mean length ± SEM of hospital stay by bacterial infection.” What is ± SEM?

*Some spacing issues thorough the manuscript

*Reviewer repeats itself in saying that the authors need to read the author’s guidelines and correctly format the references as that are not in the correct format. In addition, authors have to be more diligent with the spacing of the references as well.  For example, instructions say,

”References should be described as follows, depending on the type of work:

Journal Articles:

1.     Author 1, A.B.; Author 2, C.D. Title of the article. Abbreviated Journal Name Year, Volume, page range.”

And

“Websites:

9. Title of Site. Available online: URL (accessed on Day Month Year).”

Language issues

The numbers in the sentences are the line number from the manuscript.

*Punctuation in this sentence is wrong

 This study investigated the 17 clinical outcomes with single and combined antibiotic therapies in the face of extensive, multidrug 18 resistant and susceptible strains. (XDR, and MDR and SS) of hospital-acquired infections (HAIs).” Also what is are the following: (XDR, and MDR and SS)? Acronyms must be spelled out then used.

Difficult sentence that must be reworded or other issues

*“They were MDR (40.59%), XDR (24.75%), carbapenem-resistant enterobacteria (CRE, 27 21.78%) and susceptible (12%) strains.” For example, authors need to put the context in this sentence.

*“Monotherapy produced positive significant outcomes with E. coli 29 infections while for P. aeruginosa there were no differences between mono/combined therapies.” What is meant by *“no difference”? Do the authors mean not different in outcomes or in the therapies? It is not clear.

*“Nonetheless, combined therapy had significant effects (p<0.05) as a treatment for A. baumannii and 31 K. pneumoniae infections.” Significant effects in what exactly, numbers dropped or killed bacteria or what?  Not clear enough.

* The preva-32 lence of XDR and MDR HAIs was found to be significantly high with no association with treatment 33 type, LOHS, or outcome.” Was needs to be were. Significantly high? What p value exactly?

 *”Many risk factors are associated with HAIs, one of which is the 55 hospitalization ward, surgery and its type, as well as the patient’s underlying medical 56 conditions”  Sentence is difficult to understand with “one of which…..”  as you are not listing one but four risk factors. Remove it from and replace with “such as” for example.

*Repetition of sentences. “The antimicrobial susceptibility of the GNB isolates was also ascertained while 71 providing some insights into how patients with HADRIs are treated and the clinical 72 outcomes in the region of this study.”  This is already stated in the aim sentence lines 66-68

*Inconsistent use of GNB and Gram-negative bacterial bacteria and why is Gram- capitalized in the manuscript?

*Why is the following capitalized “Gram-positive bacteria”?

*why is the following capitalized “Gram-negative (GN)?

*”extended-spectrum beta-lactamase (ESBL)-positive strains” Hyphen is incorrectly placed here. Please correct.

*”The ID cards were inoculated with a sus-122 pension of each of the bacterial pathogens and placed into the cassette for identification.” Inaccurate way to cite a website. Please correct.

*”alive improved (AI) or alive not improved (ANI)”. This is difficult language and needs improvement. Add a simple “and” to “alive and improved” also “alive and not improved” makes the sentence and terms correct

*Inconsistent use of in Figure 3A-F or Fig. 3. Please check whole manuscript for this.

Methods

*Methods need to be written so that one can repeat the work. With this in mind all explicit data is needed; see details below.

2.2

*”The bacterial isolates of the HAIs were stored in the hos-96 pital Microbank at -80 °C from where they were retrieved, plated out on MacConkey 97 agar, cultured aerobically for 24 h at a temperature of 37 °C, and transferred to the De-98 partment of Medical Microbiology, King Faisal University.

*How many HAIs were stored? What type of isolates as in what matrices were stored and how stored; cryopreserve or? One from each 101 cases of single Gram-negative bacterial strain infections or?

*”Freshly grown, overnight 99 bacterial cultures were used for the confirmation of bacterial IDs and AST

Freshly grown how? On MacConkey agar plates or? How were the IDs ad AST done?

2.3

*Is the following already stated above already done or do authors mean the same think here. So either remove it from above or direct the sentence above to this one “Bacteria isolates were cultured aerobically on MacConkey agar at 37 ºC for 24 h. The re-115 sulting fresh bacteria colonies were used for bacteria identification and antimicrobial 116 susceptibility testing using Gram-negative (GN) ID and AST

*” sterile saline” what percentage of saline?

*” The ID cards were inoculated with a sus-122 pension of each of the bacterial pathogens and placed into the cassette for identification” how much bacteria suspension was used and inoculated how? Time temperature etc. 

2.4

*”antimicrobial susceptibility assay,” do the authors mean AST?

*”Thus, infections were categorised as multidrug resistant (MDR), extensive drug resistant 141 (XDR), or pan-drug resistant (PDR) [12].” This sentence should be placed above and earlier where the acronyms are used first.

Results

*Generally the Figures and Tables have a title and then a legend that is descriptive so that the Figure and Table can stand on its own. This means that the reader understand the Figure and Table explicitly from the legend and does not have to go to the text of the manuscript for clarity. Please do this for all the figures and tables.

*Table 1 why is there a period after .07672. and .00024., p-values are not justified and are untidy as there p-values are not centered. Table 1 needs to comment on the p-values and describe them. For example, analysis was done on what and what to give p-value of ……. Also was the frequency use or the percentage used in the analysis of p-value.  Why is there a p-value of 0.00?  Why are the two p-values .07672. and .00024. represented without a zero in the front of the value like all the other p-values?

*Figure 1 Untidy as headings are not aligned or centered. Different fonts in some cases or font size and it is pixelated so it is hard to read the text. There is inconsistency of the labels as well. Some times E. coli is used but then it is spelled out. Why the inconsistency here? In addition the y axis scales are mostly different, why? It is easier to appreciate the amounts if scaling were similar. Figure title is bold, why?  There is not legend for this

*Line 102 isolated microorganisms. Authors did not isolate the microorganisms, but ID them, right? Please reword in the entire paragraph.

*Line 194 multiple sites? Do you mean multiple departments as outlined in Table 1?

*”Figure 2 A-D illustrate the antimicrobial resistant patterns and a heatmap showing the 234 resistance profile of A. baumannii, E. coli (A), K. pneumoniae, P. aeruginosa, and other 235 Enterobacteriaceae (B) responsible for hospital-associated infections.” This statement is incorrect as 2 D does not exist in the manuscript.

It is advised that Figure 2 A is Figure 2 and Figure 2 B-C becomes Figure 3. Please make the adjustments because as it stands right now, Figure 2 split up like it is does not make sense.

*In addition, the manuscript is too confusing to have Figure 2 as it is. Where is Figure 2D anyways in the manuscript as on page 10 there is only B and C shown.

*Unconventional paragraph with one sentence. “Figure 2 A-D illustrate the antimicrobial resistant patterns and a heatmap showing the 234 resistance profile of A. baumannii, E. coli (A), K. pneumoniae, P. aeruginosa, and other 235 Enterobacteriaceae (B) responsible for hospital-associated infections.

*Inconsistency with spacing between number and letter in Figure 2A or Figure 2 A.  Check all of these.

*Figure 2B and 2C. These two panels are untidy with unreadable fonts / text, lines in some areas but in others and the label are not centered with similar spacing.

*Figure 2D is not on the page.

*Table 2. Placement is not correct in the manuscript. It should be placed when it is mentioned in the manuscript. This table is untidy as lines are not consistent in width. In fact all tables need to follow the text. Please correct placement of the tables.

Why is Unknown and Sleeve capilized in the table? Why is there a period after meningitis. In the table? Spacing of the footnote is not consistent with other tables.

*“Analysis of results showed that the age of the patients did not have any significant association with respect to type of infection, antimicrobial resistance pattern, treatment, and LOHS.” What analysis? Where is it and how can the authors state “did not have any significant association”? where is this analysis?

*“However, there was no correlation between 300age and clinical diagnosis, treatment, resistance pattern, LOHS, mortality, or no 301improvement..” Where is the correlation analysis? Why is there two periods at the end of the sentence?

*“However, 304there were no significant differences in clinical outcomes in terms of LOHS.” Where is the significant difference analysis?

*The characteristics of the infections did not show any association 311with the patients’ age, which ranged from 12 months to 87 years.” Where is this association analysis?

*Table 3. Issue with capitalized words in the table and then there isn’t. For example, why is Infected Bed Sore and Spine Infection, and Brucella capitalized but the other terms are not?

*Table 4 Line thickness is not consistent.

*Table 5. Title and table spacing is not consistent with other tables. Why is title bolded? Line thickness is not consistent in the table.

*”age had no association with infection type or mode of treatment with 322antibiotics.” Where is the association analysis?

*”(33.3%/0, followed by 323chronic otitis media (25%).” The use of the brackets is wrong here and what is meant by /0?

*Theirs ages ranged from 27 to 82 years and 333showed no correlation with clinical diagnosis or LOHS.” Where is this correlation analysis?

*”Monotherapy was significantly ef- 406 fective in the treatment of E. coli infection.” Where is this data to support this statement? And compare to what?  It seems that if a comparison was done, it has to be explicitly stated and demonstrated in the tables and figures the two comparing items. Also is the word significantly is used than the p-value should be given and explicitly seen in the figure and text. For instance, “In addition to this, the outcomes of P. 407 aeruginosa infections did not show any difference between mono and combined therapies” did not show any difference…. Where is this data? What is the p-value, etc to justify why you stated this.

*Figure 3. Figure3 (A-F). Remove the (A-F) in the title. Generally this figure is untidy. The alignment of text is off, and scales are inconsistent. The legend needs more description, especial for the statistical analysis, also what is “*” in panel A, B and D?  What methods were used to get percentage of Distribution?   Figure is pixelated. Why do font sizes differ in all the panels?

*”Double antibiotic therapy was found to be significantly (p<0.05) superior to triple 413 antibiotic therapy.” Where is this data?  Is it shown in the panel explicitly?

*”more so for A. baumannii infections” What is meant by more so for ….?

*”appeared to have produced more significant (p<0.05) effects”. What do you mean by “more significant effects” And where is this analysis explicitly in the figure?

*”combined therapy produced better results 421 in MDR bacterial infections of A. baumannii and P. aeruginosa (Figure 3D).” what is meant by better results here?

*”produced improved outcomes with monotherapy while improved outcomes 429 for other Enterobacteria needed combined therapeutic measures (Figure 3F).” what is meant by improved outcomes here?  How was it assessed?

*Figure 4. It is not correct to state this the following as it is “Indicates significant difference between E. coli and A. baumannii and K. pneumoniae infections.”  The significant differences are between E. coli and A. baumannii and then between E. coli and K. pneumoniae. Is this what you wanted to state? Why is the title and legend bolded.

*”* Indicates significant difference between E. coli” Provide the p-values in the legend too as well as in the text.

*”infections, while infections of A. baumannii and 475 other Enterobacteriaceae showed no significant differences.” Provide the p-values explicitly in the figure and text.

Discussion

*”The findings in this present report showed variations in the 490 incidence of HAIs by department/ward” The authors do mean departments, plural here.  Good to cite the data here as well like Table 1. Please do this for all the comparisons in the discussion.

*Unconventional paragraph with one sentence “With regard to the specimen type, urine and wound samples were the most common 498 here which is in line with the findings of earlier reports that cited urinary tract infections 499 (UTIs) and surgical wound infections as common HAIs [9,20,21].”  Please consider a proper paragraph structure and cite you findings here as well.

*Cite your work in this sentence “The high incidence of A. baumannii-associated HAIs seen in this investigation, 503 compared to other bacterial pathogens (E. coli, K. pneumoniae, P. aeruginosa, and other 504 enterobacteria) are contrary to those of other reports [22].

*Cite your work in this sentence “findings of A. baumannii-associated HAIs in this report simply highlights 511 the growing threat of this bacterial species in this region [26].

*Sentence structure is incorrect here. Good to make two sentences- “The widespread nature of 512 this opportunistic pathogen has been reported in hospital settings [27,28]; it has the abil-513 ity to cause a wide range of infections in immunocompromised patients [29,30], with 514 outbreaks having been reported in an adult ICU [31].

*No need for a comma in the following “it has the abil-513 ity to cause a wide range of infections in immunocompromised patients [29,30], with 514 outbreaks having been reported in an adult ICU [31].

*Cite your work in this sentence ”The resistance profile of the A. 515 baumannii HAIs to tested antibiotics reported here is worrisome.

*Cite your work here “Of the 101 HAI cases in this study, 24 521 (23.76%) died.

*Cite your work here “Although significant differences were seen here between survival and 525 mortality rates,

*”The Incidence of” Why is Incidence capitalized?

*Cite your work here “Monotherapy was preferred for E. 535 coli infections, meaning that positive outcomes were attained by single antibiotic

*Cite your work here “However, combined therapy was preferred 538 in treatments of A. baumannii and K. pneumoniae HAIs, which could be attributed to the 539 resistant nature of the bacterial pathogens (MDR, XDR CRE).

*”There is a need for more investigations 560 that would compare HADRIs in more clinical settings in critically ill patients.” More is used twice here.

*Shows or showed is used 29 times. The manuscript would benefit with the use of a thesaurus.

See above for detailed issues. 

Author Response

Kindly for the attached for your attention

Reviewer 3 Report

The authors' revision of the manuscript has significantly improved it.  Thank you to the authors for incorporating many of the suggestions.  However, the authors have not adequately responded to my major concern about the conclusions derived from Figure 3, lines 29 to 32, and lines 404 to 430.  These are critical as they prescribe a choice of treatment to different infections based on this study. 

Figure 3 is a measure of choice of mono versus combined therapy and not efficacy of treatment, however, the authors have made conclusion about the efficacy of treatment.  For example, the authors conclude combined therapy appeared to be significantly superior for A. baumannii infections.  A higher number, 16 patients were treated with combined therapy compared to 8 patients treated with monotherapy and this difference is statistically different as shown in Fig 3A.  However, from Table 2, about 44% of patients given combined therapy had a clinical outcome of alive and improved, and about 38% patients given monotherapy had a clinical outcome of alive and improved.  Based on these data, one cannot conclude that the combined therapy is significantly better.  In general, all the statistics shown for this section are from the difference in choice of treatment and not the outcome.  Similar comment applies to all other infections as well. 

Author Response

Kindly find the attached for your attention

Round 3

Reviewer 3 Report

The authors have partly addressed my major concern but seemed to have completely ignored the scientific point.  I appreciate the patience from the authors and I still raise this as I think it is critical that clinicals don't take the misinterpretation and treat patients wrongly.  I am going to again try and explain another way below, and it would be great to get a detailed rationale written response from the authors if they disagree or on what they have corrected. 

Figure 3 has nothing to do with clinical outcomes and hence, such conclusions cannot be drawn from this figure (Section 3.4 in general, for example, lines 416, 417, 427, 438).  The wording is this section is very misleading with choice of treatment mixed in with outcomes though outcomes were never used for statistical analysis.  On the other hand, the manuscript does have clinical outcomes details in terms of AI, ANI, Died; and LOHS which enable making conclusions on the outcomes of monotherapy versus combined therapy.  This data is great but is also a double edged sword.  The authors cannot ignore this data (and only focus on preferred treatment) while making conclusions in Section 3.4.  If desired, the authors could plot outcomes of monotherapy versus combined therapy for each infection and then perform statistical comparisons and draw conclusions from that with  clearly stating which data was used to make the conclusions.  I don't think these conclusions are necessary for the manuscript as the patient 'n' number might be too small and from a diverse subset to draw statistical conclusions. Alternatively, the authors could completely stay away from making any conclusions on the preferred choice of monotherapy versus combined treatment.  

Author Response

(The authors gave the same response as above.)
